# Convergent Validity of the Lower Quarter Y Balance Test Against Two-Step and Timed Up and Go Tests in Thai Older Adults with and Without Locomotive Syndrome

**DOI:** 10.3390/ijerph22040538

**Published:** 2025-04-01

**Authors:** Chadapa Rungruangbaiyok, Charupa Lektip, Jiraphat Nawarat, Eiji Miyake, Keiichiro Aoki, Hiroyuki Ohtsuka, Yasuko Inaba, Yoshinori Kagaya, Weeranan Yaemrattanakul

**Affiliations:** 1Department of Physical Therapy, School of Allied Health Sciences, Movement Science and Exercise Research Center, Walailak University, Nakhonsithammarat 80160, Thailand; chadapa.bn@wu.ac.th (C.R.); charupa.le@wu.ac.th (C.L.); nsuparoe@wu.ac.th (J.N.); 2Department of Rehabilitation, School of Nursing and Rehabilitation Sciences, Showa University, Yokohama-shi 226-8555, Kanagawa, Japan; e.miyake@nr.showa-u.ac.jp (E.M.); k.a-0525@cmed.showa-u.ac.jp (K.A.); ohtsuka@nr.showa-u.ac.jp (H.O.); inaba@nr.showa-u.ac.jp (Y.I.); kagaya@nr.showa-u.ac.jp (Y.K.); 3Department of Physical Therapy, Faculty of Medicine, Prince of Songkla University, Songkhla 90110, Thailand

**Keywords:** Y balance test, locomotive syndrome, dynamic balance, fall risk, older adults, functional mobility, timed up and go test, two-step test

## Abstract

Locomotive syndrome (LS) predisposes older adults to falls and functional dependency. In older adults with LS, the validity of the Lower Quarter Y Balance Test (YBT-LQ)—a dynamic balance assessment tool—remains unclear. This cross-sectional study aimed to assess the convergent validity of the YBT-LQ with the Two-Step and Timed Up and Go (TUG) tests and compare YBT-LQ performance between individuals with and without LS. Sixty Thai community-dwelling older adults (≥60 years) were equally divided into LS and non-LS groups and performed the YBT-LQ, Two-Step test, and TUG test. Correlation analyses and independent *t*-tests assessed relationships and between-group comparisons, respectively. The YBT-LQ exhibited moderate positive correlations with the Two-Step test (*r* = 0.366, *p* = 0.004) and moderate negative correlations with the TUG test (*r* = −0.412, *p* = 0.001). The LS group exhibited significantly lower YBT-LQ scores across all reach directions than the non-LS group (*p* < 0.05), highlighting impaired balance in individuals with LS. The YBT-LQ is a valid and reliable tool for assessing dynamic balance and postural control, as well as identifying multidirectional stability deficits in older adults, particularly those with LS. Implementing the YBT-LQ in routine geriatric evaluations could enhance early detection and targeted interventions to reduce fall risk and improve mobility in aging populations.

## 1. Introduction

As of 2022, approximately 10% of the global population is aged 65 years or older—a figure projected to increase to 16% by 2050 [1]. This demographic shift toward an aging population has significantly affected health systems and longevity worldwide. Thailand’s older adult population (defined as those aged 60 years and above) increased from 5% in 1995 to 17.1% in 2017 and is projected to reach approximately 30% by 2035 [2]. Aging is associated with a progressive decline in physiological functions across multiple systems, including the nervous system, cognition, memory, sensory perception, and musculoskeletal system [3]. These changes contribute to a range of age-related challenges, including mobility, balance, and increased risk of falls [4]. Balance and mobility impairments are particularly significant because they can lead to diminished physical function, loss of independence, and an increased likelihood of disability. This deterioration in locomotor ability is a key factor in locomotive syndrome (LS), a condition that predisposes older adults to mobility limitations and reduced quality of life [5].

LS is defined as physical, mental, and social changes that disrupt skeletal, joint, muscular, and neurological health and is commonly linked with conditions such as osteoporosis, osteoarthritis, sarcopenia, spondylosis, and neural disorders. These conditions contribute to pain, joint stiffness, muscle weakness, and diminished balance and mobility [5,6], which, in turn, elevate fall risk [7], restrict activities, interfere with daily living, reduce the quality of life [8], and create potential long-term care needs.

Early detection and prevention of locomotive syndrome rely heavily on balance assessments as they provide crucial insights into functional mobility and fall risk in older adults. With aging, maintaining the center of mass within a stable base becomes increasingly difficult because of age-related declines in musculoskeletal strength, proprioception, and postural control [8]. To evaluate these deficits, standardized tests, such as the Stand-Up and Two-Step tests, are commonly used, offering reliable measures to assess balance, lower limb function, and mobility limitations [7]. These assessments aid in early LS diagnosis and intervention, mitigate its progression, and reduce fall-related disabilities [6]. Research indicates that individuals with LS frequently experience knee and lower back pain, along with significant impairments in functional mobility. Performance in the Timed Up and Go (TUG) test, a widely used measure of walking speed and dynamic balance, has a strong correlation with LS (*r* = 0.688, *p* < 0.001) [9]. However, despite its utility, the TUG test has limitations—it primarily assesses balance in the forward direction and does not effectively differentiate multidirectional balance deficits, such as those assessed by the YBT-LQ. The test focuses on global mobility and functional performance rather than specific asymmetries or individual limb impairments.

One test that can compensate for the limitations of the TUG test is the Lower Quarter Y Balance Test (YBT-LQ). The YBT-LQ is an advanced modification of the Star Excursion Balance Test (SEBT), which initially assessed balance in eight directions but was refined to focus on three key reach directions: anterior, posteromedial, and posterolateral [10]. The SEBT and the YBT-LQ are both widely used assessments for dynamic postural control. The SEBT requires individuals to reach in multiple directions while balancing on one leg, assessing postural stability and neuromuscular control [11]. However, its administration is more complex due to variations in the number of reach directions and the need for manual measurements, which may introduce variability [12]. The YBT-LQ is a standardized refinement of the SEBT, reducing the reach directions to three key planes: anterior, posteromedial, and posterolateral. This modification enhances measurement reliability and clinical applicability while maintaining the sensitivity of the test for detecting balance deficits [11]. Additionally, the YBT-LQ incorporates a specialized testing device to minimize examiner bias and improve reproducibility, making it more practical for clinical use [12]. These specific directions were chosen based on their relevance to functional stability; the anterior and posteromedial reaches are strongly associated with chronic ankle instability, while the posterolateral reach is linked to hip extensor strength and postural control [13]. Recently, the YBT-LQ has been increasingly utilized in geriatric assessments, providing a dynamic evaluation of balance control across multiple movement planes [14]. Unlike traditional balance tests, which often assess stability in a single direction or during static stance, the YBT-LQ requires controlled single-leg reaches in three directions, offering a more comprehensive perspective on dynamic balance, postural stability, and asymmetry [15]. However, research on the YBT-LQ in older adult populations, particularly those with LS, remains lacking. While studies have confirmed the effectiveness of the YBT-LQ in measuring dynamic balance, its validity relative to established assessments, such as the TUG and Two-Step test, in older adults with LS has not yet been investigated. Furthermore, no study has explored the performance differences in the YBT-LQ, TUG test, and Two-Step test between older individuals with and without LS, leaving a critical gap in our understanding of how accurately these tests assess balance impairments unique to this condition.

In addition to balance assessments, effective fall prevention strategies include structured exercise programs aimed at improving balance, coordination, and mobility. Such interventions typically focus on lower limb strengthening, proprioceptive training, and postural control enhancement. Evidence suggests that multidirectional training, including tasks that challenge dynamic stability, can be particularly beneficial in reducing fall risk and improving overall functional mobility in older adults [16,17,18,19,20]. These strategies should be integrated into routine physical therapy and rehabilitation programs for individuals at risk of locomotor decline.

Therefore, this study aimed to address this vital knowledge gap by clarifying the application of the YBT-LQ in older adults with LS, potentially guiding clinical assessments and interventions to enhance balance and mobility in this vulnerable population. To this end, this study had two primary objectives: (1) to investigate the convergent validity of the YBT-LQ against the Two-Step test and TUG test in Thai older adults aged ≥60 years, both with and without LS; and (2) to compare the YBT-LQ, TUG test, and Two-Step test scores between older participants with and without LS to validate the capability of the YBT-LQ to assess dynamic balance.

## 2. Materials and Methods

### 2.1. Study Design

This cross-sectional analytical study was designed to evaluate the convergent validity of the YBT-LQ against the Two-Step and the TUG tests in older individuals with and without LS.

### 2.2. Ethical Considerations

This study was approved by the ethics committee of Walailak University and followed the principles of the Declaration of Helsinki (approval number: WUEC-24-083-01), granted on 8 March 2024. Data collection commenced on 13 March 2024. During the data analysis process, an amendment was submitted to include additional researchers, which was reviewed and approved by the Ethics Committee on 4 November 2024. This amendment did not alter the study protocol, data analysis procedures, or ethical compliance.

### 2.3. Participants

Our study included 60 participants, equally divided into two groups: 30 individuals with LS and 30 individuals without LS. The target population for this study included Thai male and female older adults residing in Nakhon Si Thammarat Province, Thailand.

Participants eligible for the study were Thai nationals aged 60 years or older who were able to walk independently without assistive devices and stand on one leg for at least 10 s on both sides. Additionally, they were required to have the ability to read and communicate in Thai.

Participants were excluded from the study if they were unable to successfully complete any of the required assessments. Individuals who had undergone spinal or lower extremity surgery affecting mobility within the past three months were not eligible. Those with lower limb amputation in any case, moderate to severe pain (greater than 5/10 on a pain scale) affecting movement, or a diagnosed neurological disorder were also excluded. Furthermore, individuals with dementia, heart failure, or uncontrolled cardiovascular, respiratory, vascular, or metabolic disorders were not included. Participants experiencing dizziness or vertigo were also excluded from the study to ensure their safety during balance assessments.

### 2.4. Tools and Instruments

The YBT-LQ utilizes a specialized apparatus consisting of a standing platform with three PVC pipes attached in the direction of reach: anterior reach, posteromedial reach, and posterolateral reach (Figure 1). The posterior pipes were positioned 135° from the anterior reach pipe, with a 45° angle between the posteromedial and posterolateral pipes [21].

The apparatus was equipped with directional markers to ensure consistent and accurate measurements, and standard measuring tape was attached to the pipes for precise distance recording. Each pipe was marked at 5 mm intervals to facilitate measurement. The participants used their feet to push the box along the pipe, with the farthest distance reached set as the standard, ensuring that the box remained above the measuring line while the value was recorded. Upon completion of the test, each individual’s reach distance was accurately assessed.

### 2.5. Procedure

All procedures were carried out by trained investigators after obtaining informed consent from the participants. All test results were manually recorded on structured data collection forms and later digitized for analysis. The collected data were securely stored and managed using a dedicated database to ensure confidentiality and accuracy. To screen for dementia, the Mini-Cog test was administered prior to the physical performance assessments. This screening tool evaluates short-term memory and executive function, ensuring that participants can accurately follow instructions during the testing procedures. Demographic and clinical information, including age, sex, and co-morbidity, was collected through interviews.

#### 2.5.1. Baseline Anthropometric Measurements

Height was measured using a wooden stadiometer (SPORTLAND Height wooden Fold Measuring; Sport Land Corporation, Thailand). Body weight was measured using a digital scale (Tanita HD-380; Tanita Corporation, Tokyo, Japan). Limb length was measured using a standard measuring tape (measured from the anterior superior iliac spine to the distal medial malleolus).

#### 2.5.2. Two-Step Test

The Two-Step test was used to assess locomotor function, and participants were categorized based on their performance. According to Ishibashi (2018) [22], a Two-Step test score of less than 1.3 indicates the presence of Locomotive Syndrome (LS). The test procedure involved aligning both feet at the starting line, after which participants were instructed to take the longest possible two steps forward without losing balance. Each participant performed the test three times, and the best distance was recorded. The final score was then normalized to the participant’s height to ensure an accurate assessment of step length relative to body proportions.

#### 2.5.3. YBT-LQ Test

Participants were shown an instructional video and given practice trials to familiarize themselves with the test protocol. Standing barefoot on the testing platform, the participants performed single-leg reaches in three directions: anterior, posteromedial, and posterolateral. Each direction was tested three times per leg, alternating between the left and right leg, with a 10 s rest between trials and a 20 s rest between directions. The reach distances were averaged and normalized to limb length. The trials were repeated if the participants lost balance, moved their support foot, or failed to return to their starting position.

#### 2.5.4. TUG Test

Starting from a seated position on a chair, the participants were instructed to stand up, walk 3 m as quickly as possible without running, turn around a cone, and return to the chair. The test was performed thrice, and the fastest time was recorded for analysis.

### 2.6. Sample Size Calculation

The sample size for this study was calculated as 60 participants using proportional stratified random sampling. Participants included 30 older adults with and 30 without Locomotive Syndrome, residing in Nakhon Si Thammarat Province. Each group consisted of 15 males and 15 females. The sample size was determined using the G*Power program based on specific study objectives.

To assess the convergent validity of the YBT-LQ compared with the TUG test in older adults with Locomotive Syndrome, the statistical test used was a correlation analysis with a bivariate normal model. The parameters included a significance level (α) of 0.05, power (1 − β) of 0.8, and a correlation coefficient (r) of 0.5. The correlation value was derived from a previous study by Cody L. et al. (2019) [14] on the relationship between the YBT-LQ and TUG in healthy older adults. The calculation indicated that 29 participants per group were required.

To compare YBT-LQ performance between older adults with and without Locomotive Syndrome, an independent means comparison test was applied. The statistical test measured the difference between two independent means (two groups) with a significance level (α) of 0.05, power (1 − β) of 0.8, and effect size (d) of 0.8. The effect size was obtained from a previous study by Ahmad A. (2010) [23]. The calculation indicated that 52 participants were needed, with 26 participants per group.

This sample size determination follows statistical principles and ensures sufficient power to address both study objectives.

### 2.7. Statistical Analysis

IBM SPSS Statistics version 25 (IBM, Armonk, NY, USA) was used for data entry, editing, and statistical analyses. Statistical significance was set at *p* < 0.05. Descriptive statistics were used to summarize the participants’ demographic and physical characteristics, including age, weight, height, body mass index (BMI), and limb length. Data normality was assessed using the Kolmogorov–Smirnov test. For normally distributed data, independent sample *t*-tests were used to compare YBT-LQ scores and other outcome measures between participants with and without LS. For non-normally distributed data, the Mann–Whitney U test was applied.

Correlation analyses were conducted to evaluate the convergent validity of the YBT-LQ against the Two-Step and TUG tests. Pearson’s correlation coefficient was used for parametric data, whereas Spearman’s was applied to non-parametric data. The strength and direction of the correlations were interpreted to determine the relationship between YBT-LQ performance and that of the other tests.

## 3. Results

### 3.1. Participant Characteristics

The demographic and physical characteristics of the participants are summarized in Table 1. Participants were divided into LS and non-LS groups. No significant differences were observed in age (67.2 ± 4.6 and 69.3 ± 6.2 years, *p* = 0.156) or height (159.7 ± 7.7 and 160.1 ± 10.2 cm, *p* = 0.790) between the non-LS and LS groups, respectively. The LS group exhibited a slightly higher BMI (25.5 ± 4.9 kg/m^2^) than the non-LS group (23.6 ± 4.3 kg/m^2^), although this difference was not statistically significant (*p* = 0.060).

### 3.2. Performance Comparisons

Significant differences in the performance metrics were identified between the LS and non-LS groups (Table 1).

Two-Step test: The non-LS group outperformed the LS group (1.34 ± 0.10 vs. 1.06 ± 0.13, *p* < 0.001), indicating better dynamic stability in individuals without LS.

TUG test: The LS group required significantly more time to complete the TUG test (7.2 ± 1.0 s) than the non-LS group (6.6 ± 0.8 s, *p* = 0.014), reflecting reduced mobility.

YBT-LQ: The non-LS group achieved significantly higher scores (81.2 ± 10.4) than the LS group (70.1 ± 11.3, *p* < 0.001), highlighting poorer dynamic balance in participants with LS.

### 3.3. Correlation Results

Figure 2 and Figure 3 illustrate the relationships between the YBT-LQ, Two-Step test, and TUG test results. In addition, Table 2 summarizes the correlation coefficients between the YBT-LQ and the Two-Step and TUG tests. A moderate positive correlation was observed between the Two-Step test and YBT-LQ scores (*r* = 0.366, *p* = 0.004). A moderate negative correlation was noted between the TUG test and YBT-LQ scores (*r* = −0.412, *p* = 0.001), indicating that lower balance performance on the YBT-LQ was associated with slower mobility.

### 3.4. Directional Reach Analysis

As depicted in Table 3, significant differences in reach distances were observed in the YBT-LQ in the anterior, posteromedial, and posterolateral directions for both legs. Across all directions, the non-LS group exhibited greater reach distances than the LS group (*p* < 0.05).

## 4. Discussion

This study investigated the convergent validity of the YBT-LQ against the Two-Step and TUG tests in older Thai adults with and without LS. The findings demonstrated that the YBT-LQ outcomes significantly correlated with the outcomes of both functional mobility tests, reinforcing its utility as an assessment tool for dynamic balance in aging populations. Furthermore, individuals with LS exhibited significantly lower YBT-LQ scores across all three reach directions, highlighting the effects of LS on multidirectional postural control. These results contribute to a growing body of evidence supporting the use of the YBT-LQ as a reliable tool for evaluating balance impairment and fall risk in older adults [24,25].

The moderate positive correlation between the YBT-LQ and Two-Step test outcomes (*r* = 0.366, *p* = 0.004) and the moderate negative correlation with the TUG test (*r* = −0.412, *p* = 0.001) suggest that these functional tests partly share underlying neuromuscular control mechanisms with the YBT-LQ. However, these correlations were only moderate, indicating that the YBT-LQ captures additional aspects of balance beyond those assessed by the Two-Step and TUG tests [26]. Specifically, the YBT-LQ incorporates multidirectional challenges, such as lateral and posterior reach components, which are less represented in the linear movements of the Two-Step and TUG tests. This finding aligns with the idea that YBT-LQ may assess dynamic balance components not fully captured by traditional mobility tests [15,16]. Postural control involves sensory integration, motor responses, and anticipatory adjustments to maintain balance. The YBT-LQ measures multidirectional reach distances, reflecting distinct aspects of balance compared to the TUG and Two-Step tests. The TUG evaluates forward mobility and transitions [27,28], while the Two-Step assesses dynamic forward stepping [5,9]. By capturing lateral and posterior stability—critical for activities like side-stepping—the YBT-LQ addresses balance dimensions overlooked by TUG and Two-Step tests [15,29]. This distinction reinforces that YBT-LQ assesses postural control elements beyond those measured by TUG and Two-Step tests.

Beyond establishing convergent validity, this study also revealed significant differences in YBT-LQ performance between individuals with and without LS. Participants with LS exhibited significantly shorter reach distances across all three movement directions (anterior, posteromedial, and posterolateral) for both legs. These findings are consistent with those of previous studies, indicating that LS is associated with reduced muscle strength, proprioceptive control, and neuromuscular efficiency [5,6]. Anterior reach, which relies heavily on ankle dorsiflexion for foot mobility and quadricep control for knee stability, was notably reduced in individuals with LS, indicating potential impairments in forward-directed stability [30,31]. Likewise, deficits in the posteromedial and posterolateral directions highlight challenges in hip stabilization and lateral postural control, both of which are critical for maintaining balance during weight-shifting activities [15,21]. These results reinforce the role of LS as a progressive condition that compromises not only mobility but also the ability to maintain postural stability in multiple planes of motion. This is supported by Rezaei et al. [32], who demonstrated significant age-related declines in the unipedal stance time and increased center of pressure movement, indicating worsening neuromuscular control and postural instability with aging.

The significant differences in YBT-LQ performance between the LS and non-LS groups have important clinical implications. Given that LS is an early-stage musculoskeletal disorder that predisposes older adults to a decline in mobility and an increased risk of falls, the ability to detect subtle balance impairments before severe functional limitations arise is crucial [33]. The multidirectional nature of the YBT-LQ makes it particularly suitable for identifying specific areas of balance dysfunction, allowing clinicians to design targeted interventions [14,29]. For example, anterior reach deficits may indicate the need for ankle dorsiflexion and quadriceps strengthening exercises, while posterolateral reach limitations may suggest the need for hip stability and lateral movement training [15,34]. These targeted rehabilitation strategies can be integrated into physical therapy programs to address specific postural deficits and reduce the risk of falls in older adults with LS [35].

Furthermore, the ability of the YBT-LQ to assess asymmetries between the limbs provides additional clinical value. Previous research has shown that older adults with mobility impairment often exhibit greater side-to-side differences in dynamic balance, which can contribute to gait instability and an increased likelihood of falls [21,36,37]. In this study, LS participants demonstrated reduced reach distances across all directions on the YBT-LQ compared to non-LS participants, reflecting multidirectional postural control deficits. Despite moderate correlations with TUG and Two-Step results, the YBT-LQ offers complementary insights, guiding targeted interventions. For example, anterior deficits may signal quadriceps weakness, while posterolateral deficits suggest hip instability [15,29]. Integrating the YBT-LQ into clinical assessments can facilitate personalized balance training programs, addressing specific postural impairments and reducing fall risk [14,34]. Moreover, interventions should emphasize multidirectional balance enhancement to address LS-related postural impairments. As aging progresses, older adults commonly experience declines in postural stability, proprioception, and muscle strength, which contribute to increased fall risk and mobility impairments [38,39]. These factors are particularly relevant in individuals at risk for Locomotive Syndrome (LS), a condition characterized by progressive deterioration of musculoskeletal function leading to mobility limitations [40].To mitigate these risks, interventions should emphasize multidirectional balance enhancement to address age-related postural impairments, integrating evidence-based rehabilitation approaches. Recent studies suggest that a combination of structured strength training, proprioceptive exercises, and multimodal balance training can effectively improve postural stability and mobility in older adults, supporting functional independence and reducing fall risk [41,42]. Given that LS is associated with declining physical function, incorporating these interventions may play a preventive role in maintaining mobility and delaying the onset of severe movement limitations in aging populations.

Additionally, the LS group had a slightly higher BMI (*p* = 0.060), which may have influenced their lower YBT-LQ scores, as higher BMI is known to impair balance and mobility [43,44,45]. Future research should further investigate the combined impact of BMI and limb asymmetry on YBT-LQ performance and fall risk. Additionally, interventions should emphasize multidirectional balance enhancement to address LS-related postural impairments.

Although the findings of this study support the validity and clinical utility of the YBT-LQ in older adults with LS, several limitations must be acknowledged. First, its cross-sectional design precluded any causal inference regarding the relationship between YBT-LQ performance and functional mobility decline. A longitudinal approach would be beneficial for determining whether YBT-LQ scores can predict future mobility impairment and fall incidence in populations with LS. Second, although a sample size of 60 participants provided sufficient statistical power to detect significant associations, larger-scale studies with diverse populations are needed to enhance the generalizability of the findings. Third, this study focused on community-dwelling older adults who could walk independently without using assistive devices. Future research should examine the applicability of the YBT-LQ in more functionally impaired populations, including those at higher risk of falls or residing in long-term care facilities. In addition, while medication use and fall history are known to influence fall risk in older adults [46,47,48], our study did not aim to assess fall risk directly. Instead, our primary objective was to examine the convergent validity of the YBT-LQ in comparison to the Two-Step test and TUG test in older adults with and without Locomotive Syndrome (LS). Therefore, we did not collect or analyze data on medication usage or fall history as part of the participant characteristics. Future research could explore these factors to better understand their impact on dynamic balance assessments and mobility outcomes.

Despite these limitations, the findings have several implications for clinical practice and future research. The moderate correlations between the YBT-LQ, Two-Step test, and TUG test outcomes support the use of the YBT-LQ as a complementary assessment tool for balance and mobility in aging populations. Moreover, the observed differences between the LS and non-LS groups highlight the need for early screening and intervention strategies targeting balance impairment in individuals at risk of LS progression. Given the growing global burden of musculoskeletal disorders in aging populations, incorporating the YBT-LQ into routine physical therapy assessments could aid in the early identification of individuals at risk for mobility decline, allowing for timely intervention to prevent falls and maintain independence.

## 5. Conclusions

In conclusion, this study demonstrates that the YBT-LQ is a valid measure of dynamic balance in older adults, exhibiting significant correlations with established mobility assessments, such as the Two-Step and TUG tests. Additionally, the significant differences in YBT-LQ performance between individuals with and without LS underscore its potential as a diagnostic tool for identifying balance deficits in populations at risk of mobility decline. Future research should focus on longitudinal studies to determine the predictive validity of the YBT-LQ for fall risk as well as interventional studies to assess its utility in guiding balance training programs for older adults with LS.

## Figures and Tables

**Figure 1 ijerph-22-00538-f001:**
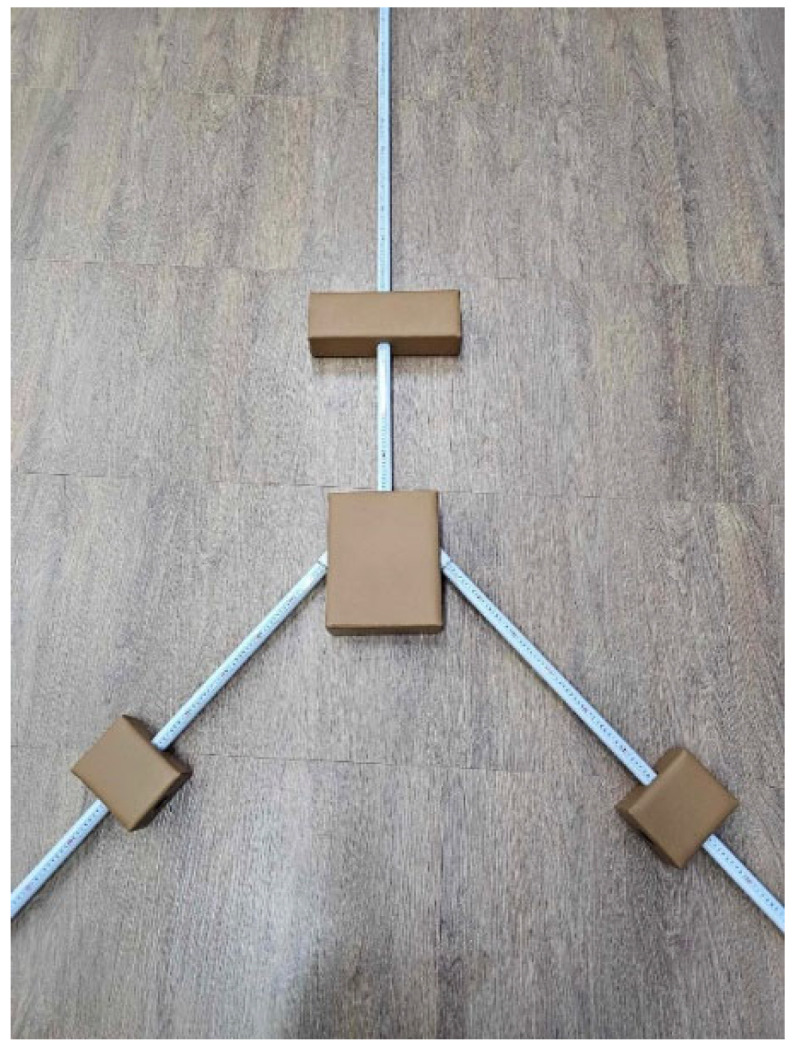
Lower Quarter Y Balance Test Apparatus.

**Figure 2 ijerph-22-00538-f002:**
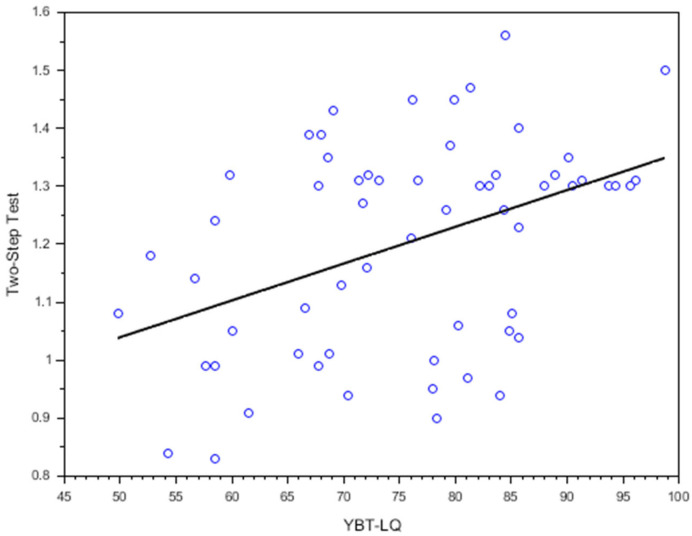
Scatterplots with regression lines illustrating correlations between the YBT-LQ and two-step test. Each circle represents an individual participant’s data point. The regression line indicates the trend and strength of the relationship between the two variables.

**Figure 3 ijerph-22-00538-f003:**
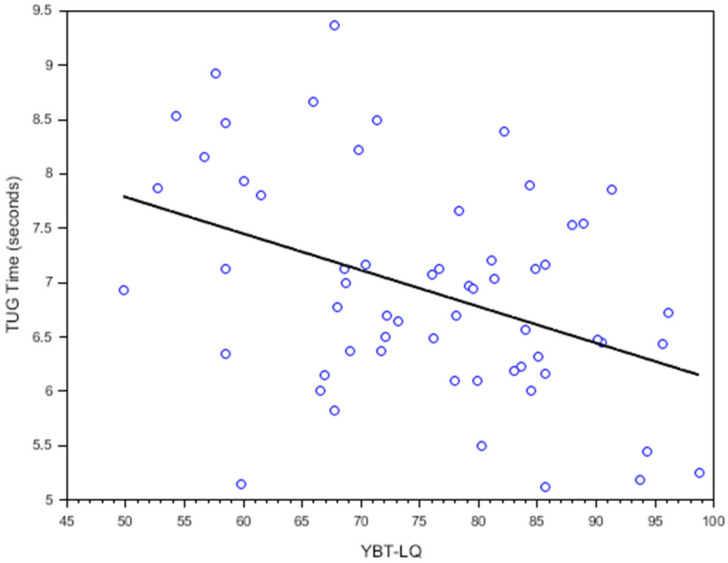
Scatterplots with regression lines illustrating correlations between the YBT-LQ and TUG test = timed up and go test. Each circle represents an individual participant’s data point. The regression line indicates the trend and strength of the relationship between the two variables.

**Table 1 ijerph-22-00538-t001:** Characteristics and performance measures of participants (N = 60).

Characteristics	Non-Locomotive Syndrome (*n* = 30)		Locomotive Syndrome (*n* = 30)		*p*-Value
	Mean ± SD	Range	Mean ± SD	Range	
Age (years)	67.2 ± 4.6	61–77	69.3 ± 6.2	60–87	0.156 ^a^
Weight (kg)	60.1 ± 10.4	48.0–99.5	63.8 ± 12.7	40–89	0.226 ^a^
Height (cm)	159.7 ± 7.7	147–185	160.1 ± 10.2	143–185	0.790 ^b^
BMI (kg/m^2^)	23.6 ± 4.3	17.6–38.9	25.5 ± 4.9	16.96–35.4	0.060 ^b^
Two-Step test	1.3 ± 0.1	1.0–1.6	1.1 ± 0.1	0.8–1.3	<0.001 *^,b^
TUG test (sec)	6.6 ± 0.8	5.2–8.5	7.2 ± 1.0	5.1–9.4	0.014 *^,a^
YBT-LQ (%)	81.2 ± 10.4	59.8–98.7	70.1 ± 11.3	49.8–85.7	<0.001 *^,a^

^a^ independent sample *t*-test; ^b^ Mann–Whitney U test; * statistically significant (*p* < 0.05). BMI: body mass index; TUG: Timed Up and Go; YBT-LQ: Lower Quarter Y Balance Test.

**Table 2 ijerph-22-00538-t002:** Correlations between the YBT-LQ and the comparison outcome measures (N = 60).

		Two-Step Test	TUG Test
YBT-LQ	Correlation coefficient	0.366	−0.412
	*p*-value	0.004 *^,a^	0.001 *^,b^

^a^ Spearman’s rho; ^b^ Pearson’s correlation; * Statistically significant (*p* < 0.05). TUG: Timed Up and Go; YBT-LQ: Lower Quarter Y Balance Test.

**Table 3 ijerph-22-00538-t003:** Comparison of YBT-LQ outcomes for each side and direction between the non-LS and LS groups.

Side	Direction	Non-Locomotive Syndrome	Locomotive Syndrome	*p*-Value
		Mean ± SD	Mean ± SD	
Left leg	Anterior	67.7 ± 6.9	59.7 ± 11.4	0.002 *
	Posteromedial	82.9 ± 11.3	71.0 ± 12.4	<0.001 *
	Posterolateral	85.6 ± 13.4	70.1 ± 14.7	<0.001 *
Right leg	Anterior	69.2 ± 8.8	60.1 ± 8.9	<0.001 *
	Posteromedial	80.8 ± 9.8	73.4 ± 12.9	0.015 *
	Posterolateral	83.8 ± 13.5	72.4 ± 12.7	0.001 *

*p*-value determined using independent sample *t*-tests; * statistically significant (*p* < 0.05). YBT-LQ: Lower Quarter Y Balance Test; LS: locomotive syndrome; SD: standard deviation.

## Data Availability

The data presented in this cross-sectional study are not publicly available due to ethical restrictions and participant confidentiality. Data may be made available from the corresponding author upon reasonable request and with approval from the Walailak University Ethics Committee.

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
