# Peer review of "Convergent Validity of the Lower Quarter Y Balance Test Against Two-Step and Timed Up and Go Tests in Thai Older Adults with and Without Locomotive Syndrome"

_ijerph, 2025, doi:10.3390/ijerph22040538_

Round 1
Reviewer 1 Report
Comments and Suggestions for Authors
Dear Authors
Above all the intention of the research in the manuscript, I only have two senteces that is possible, please add.
To prevent the risk of falls and to provide all the significance about this tests to evaluate balance and mobility, what are the main strategies to improve it: To improve the balance, the coordination, the mobility?
Also, test could improve the reference to evalute better but for what?
May be the authors should add more information in introduction and discussion above physical exercise, strategies to prevent falls..
Comments on the Quality of English Language
Nothing to declare
Reviewer 2 Report
Comments and Suggestions for Authors
Dear authors,
The manuscript entitled “Convergent Validity of the Lower Quarter Y Balance Test against Two-Step and Timed Up and Go Tests in Thai Older Adults With and Without Locomotive Syndrome” is gaining rapid interest in medical care facilities, as well as in daily life or any facility focused on balance, gait, and risk of fall in the elderly.
While the topic is interesting, there are some aspects that should be addressed:
- I suggest to also add some more relevant articles in your references regarding the locomotive syndrome, risk of falls in the elderly, as well as on static and dynamic balance, and Star Excursion Balance Test.
- Please add more information regarding the Lower Quarter Y Balance Test in comparison to the Star Excursion Balance Test.
- If possible (if data is already available), I would highly suggest to add in the “Characteristics and performance measures of participants (N = 60) Table” data regarding the medication (number) and history of falls, since it has been known that the greater is the number of medication used, the greater is the risk for falls.
- Please state more clearly the inclusion and exclusion criteria in the manuscript to be easier to follow information.
- I highly recommend to add more specific details for the methodology of gathering all the information.
- Did the participants have any specific musculoskeletal injuries or other conditions which could have interfered with the dynamic balance and postural control?
- Why didn’t you take into consideration a comparison with the Star Excursion Balance Test?
- Did you also use a computerized quantitative analysis for dynamic balance and gait to compare with the results? This would help with a predictive analysis.
- I highly suggest to compare your results with more recent studies, including the rehabilitation interventions for those with static and dynamic balance anomalies based on these tests you also used, as well as vestibular interventions, vibration therapy or any other type of therapies for those individuals afflicted with locomotive syndrome.
Good luck!
Round 2
Reviewer 2 Report
Comments and Suggestions for Authors
Dear Authors,
Thank you for incorporating the suggestions in your manuscript. I believe that now it can fully come towards the readers with more clear and strong information.
Good luck!